# Federated learning for millimeter-wave spectrum in 6G networks: applications, challenges, way forward and open research issues

Faizan Qamar[1], Syed Hussain Ali Kazmi[1], Maraj Uddin Ahmed Siddiqui[2], Rosilah Hassan[1] and Khairul Akram Zainol Ariffin[1]

[1] Center of Cyber Security, Faculty of Information Science and Technology, Universiti Kebangsaan Malaysia, Bangi, Selangor, Malaysia
[2] James Watt School of Engineering, University of Glasgow, Glasgow, United Kingdom

## ABSTRACT

The emergence of 6G networks promises ultra-high data rates and unprecedented connectivity. However, the effective utilization of the millimeter-wave (mmWave) as a critical enabler of foreseen potential in 6G, poses significant challenges due to its unique propagation characteristics and security concerns. Deep learning (DL)/machine learning (ML) based approaches emerged as potential solutions; however, DL/ML contains centralization and data privacy issues. Therefore, federated learning (FL), an innovative decentralized DL/ML paradigm, offers a promising avenue to tackle these challenges by enabling collaborative model training across distributed devices while preserving data privacy. After a comprehensive exploration of FL enabled 6G networks, this review identifies the specific applications of mmWave communications in the context of FL enabled 6G networks. Thereby, this article discusses particular challenges faced in the adaption of FL enabled mmWave communication in 6G; including bandwidth consumption, power consumption and synchronization requirements. In view of the identified challenges, this study proposed a way forward called Federated Energy-Aware Dynamic Synchronization with Bandwidth-Optimization (FEADSBO). Moreover, this review highlights pertinent open research issues by synthesizing current advancements and research efforts. Through this review, we provide a roadmap to harness the synergies between FL and mmWave, offering insights to reshape the landscape of 6G networks.

# INTRODUCTION

The current fifth generation (5G) cellular networks are the pillar radio service provider that enables revolutionary applications and activities. It greatly expands mobile services and provides superior network performance, not only in the telecommunication sector but also in many other arenas such as the industrial (*Wollschlaeger, Sauter & Jasperneite, 2017*), healthcare (*Catarinucci et al., 2015*), educational (*Dake & Ofosu, 2019*), defense (*Lee, Baek & Choi, 2021*), and automobile sectors (*Papadimitratos et al., 2009*), *etc.* It has also been observed that current spectrum resources and high power dissipation of diverse technological products would not be able to satisfy the upcoming user data provision and

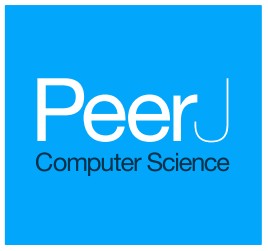

Corresponding author
Faizan Qamar,
faizanqamar@ukm.edu.my

**Table 1 Key performance difference between 4G, 5G, and 6G.**

| Performance metric | 4G | 5G | 6G |
|---|---|---|---|
| Energy efficiency (energy/bit) | ~90% more | 90% or less | 95% or less |
| Latency (ms) | 20–30 | Less than 1 | Up to 0.1 |
| Maximum spectrum efficiency (bps/Hz) | 15 | 30 | 100 |
| Connectivity (smart devices/km$^2$) | 10–100 K | More or less 1 million | Approx. 5–10 million |
| Available spectrum (GHz) | Sub-6 | Up to 300 | Up to 3,000 |
| Mobility (m/hr) | 200–250 | 300–400 | 600 |
| AI/ML | Not used | Partial | Fully |
| Maximum throughput (Gbps) | 0.5–0.6 | ~10 | Up to 1,000 or more |
| Environment | MIMO | M-MIMO | Intelligent surfaces |
| Satellite integration | No | No | Fully |

energy management (*Siddiqui et al., 2022*). These constraints demand modification in conventional schemes, innovative designs, new protocols, and robustness in cellular data packet transmission techniques. The performance difference between the previous fourth-generation (4G), current 5G, and upcoming 6G is demonstrated in Table 1 (*Chataut & Akl, 2020*). In the ever-evolving landscape of wireless communication, the advent of 6G networks holds the promise of revolutionizing data rates and connectivity to an unparalleled degree. The transition from 5G to 6G communication must be carried through several technological shifts and backward compatibilities. At the heart of this transformation lies the millimeter-wave (mmWave), a precious resource with the potential to unlock the full capabilities of 6G networks.

However, the effective harnessing of the mmWave spectrum comes with its own challenges, rooted in its distinctive propagation characteristics and intricate privacy considerations. Innovative approaches are imperative to address these challenges and ensure the seamless integration of mmWave technology into the fabric of 6G networks. Federated learning (FL), emerging as a decentralized and collaborative machine learning paradigm, offers a potential avenue to overcome the hurdles associated with mmWave spectrum utilization. By enabling the collective training of machine learning models across a distributed array of devices, FL ensures data privacy preservation while achieving remarkable levels of collaboration. Researchers are pursuing exploration of the potential applications of FL techniques to establish efficient mmWave spectrum usage within the context of 6G networks, especially through exploiting the MIMO technology. The technical intricacies of deploying FL in the context of mmWave spectrum usage are multifaceted and demand careful consideration. One of the foremost challenges arises from the inherent communication bandwidth limitations within the mmWave spectrum. Addressing this limitation requires innovative solutions to facilitate effective model updates while minimizing the overhead imposed by data transmission. Moreover, the energy efficiency of devices engaged in FL-based spectrum management becomes crucial in resource-constrained mmWave communication systems. Synchronization emerges as a

pivotal issue when dealing with the distributed nature of FL in a mmWave-enabled environment. The propagation characteristics of the mmWave spectrum introduce unique delays and synchronization challenges that necessitate tailored synchronization mechanisms to ensure coherent model aggregation.

In addition to the technical intricacies, privacy and security considerations take center stage in utilizing mmWave spectrum (*Kazmi et al., 2023a*). The very nature of distributed machine learning (ML) in FL aligns with the imperative to uphold data privacy. However, the privacy challenges intrinsic to mmWave-based communications introduce novel vulnerabilities that demand specialized attention. These challenges include eavesdropping, signal interception, and potential attacks on communication nodes. Herein lies the potential of FL to act as a robust countermeasure by ensuring that sensitive data remains localized while allowing for collective intelligence to be harnessed. The optimization of FL algorithms tailored to the unique demands of mmWave spectrum allocation stands as an important area of investigation. Enhancing the collaborative nature of FL across diverse devices within a mmWave network ecosystem can unlock new dimensions of efficiency and efficacy. Robust security frameworks that integrate FL techniques have the potential to fortify mmWave communications against emerging threats. Similarly, future ultra-dense wireless networks are expected to provide extremely high data rates with minimal radio frequency (RF) transmission losses to services like ultra-high-definition (UHD) applications, immersive media, and ultra-fast mobile vehicular communication. Still, they are heavily susceptible to geographical and environmental challenges. This would result in a high symbol error rate (SER), poor signal-to-interference-plus noise ratio (SINR) level, and ultimately substandard overall quality-of-experience (QoE), which are entirely unacceptable (*Kazmi et al., 2023b*; *Hindia et al., 2019*). Therefore, this review article offers insight into FL technology and its prospects for mmWave in 6G networks by discussing different aspects and their relevant areas (*Jha & Singh, 2013*). This article identifies potential areas in the subject domain by amalgamating current advancements and ongoing research endeavors. Further, this article comprehensively covers challenges associated with FL enabled mmWave spectrum utilization. Through carefully examining technical intricacies, privacy concerns, and challenges, this review strives to lay the future research directions in the subject domain for a transformative era in wireless networking.

## Rationale and targeted audience

The study is essential for several reasons. Firstly, as 6G networks promise ultra-high data rates and connectivity, effectively utilizing the millimeter-wave spectrum is crucial for realizing their full potential. However, the unique propagation characteristics and security concerns of mmWave pose significant challenges. Traditional DL/ML approaches have limitations such as centralization and data privacy issues. Therefore, exploring FL as a decentralized paradigm becomes imperative, offering a solution to collaborate on model training while preserving data privacy. This study aims to address these challenges, identify specific applications, propose solutions, and highlight open research issues to pave the way for integrating FL and mmWave in 6G networks.

**Table 2 A comparative analysis of related previous surveys and scope of this article.** Annotations: "√" indicates that concepts are covered comprehensively, "0" indicates that scope is partially covered, "X" indicates that scope is not covered.

| Ref. | Year | FL 6G | mm Wave | FL mmWave | 6G | Contributions | Limitations |
|------|------|-------|---------|-----------|-----|---------------|-------------|
| Li et al. (2020) | 2020 | 0 | X | X | 0 | • Review on FL application in mobile device and industrial domain | • Do not include mmWave applications |
| Lim et al. (2020) | 2020 | 0 | X | X | 0 | • Analyzes FL integration with mobile edge computing | • Lacks the discussion on mmWave based wireless communication. |
| Wahab et al. (2021) | 2021 | √ | X | X | X | • Exploration of evolving FL in communication and networking | • Lacks discussion on mmWave specific FL application and concepts in 6G networks |
| Boulogeorgos et al. (2021) | 2021 | √ | 0 | X | 0 | • Explores the applications of AI in THz wireless | • Does not fully cover mmWave and FL enabled 6G networks |
| Pham et al. (2022) | 2022 | √ | 0 | X | √ | • Analyzes utilization of MEC to for integration of FL in context of 6G networks | • Does not relate the discussion with mmWave unitization in 6G networks |
| Al-Quraan et al. (2023) | 2023 | √ | 0 | 0 | √ | • Explore cutting-edge FL applications within wireless technologies | • Partially discuss areas related to mmWave spectrum |
| Duan et al. (2023) | 2023 | √ | 0 | X | 0 | • Provides an in-depth examination of technologies that integrate FL and edge computing | • Does not discuss concepts for mmWave communication in FL enabled 6G networks |
| Driss et al. (2023) | 2023 | √ | 0 | 0 | √ | • Explores the integration of FL across the entire protocol stack in the 6G technology | • Only brief discussion on mmWave utilization in FL enabled 6G network |
| Xiao et al. (2024) | 2024 | 0 | 0 | 0 | 0 | • Analyzes various scenarios in FL over the air including, MIMO and RIS | • Only partially discussed with respect to RIS integration with FL |
| Lee et al. (2024) | 2024 | 0 | 0 | 0 | 0 | • Analyzes FL concepts in 5G and Beyond network including network management, network core, network access | • mmWave communication is discussed only partially in LTE communication domain. |

The targeted audience for this study includes researchers, engineers, and practitioners in the fields of telecommunications, wireless networking, machine learning, and artificial intelligence. Specifically, professionals involved in the development and deployment of 6G networks. Moreover, this study is particularly relevant for the research community interested in leveraging advanced techniques like FL to enhance network performance and security. Additionally, policymakers and stakeholders in the telecommunications industry seeking insights into the future directions of network technology and its implications on data privacy and security would benefit from the findings and recommendations presented in this review.

## Related literature and their limitations

Recently, there has been a remarkable surge of research publications/reviews/surveys on the FL enabled 6G communication networks, spanning all the layers in the emerging

network architecture. Table 2 summarizes the discussion on previous research works that are related.

The authors in Li et al. (2020) provide a detailed review of FL applications in mobile device and industrial domains but do not include mmWave applications. Similarly, the study Lim et al. (2020) is a comprehensive survey on FL integration with mobile edge computing but lacks a discussion on mmWave based wireless communication with FL. The study Wahab et al. (2021) offers a comprehensive understanding of federated learning's intricacies, followed by exploring its evolving role and prospects in communication and networking. However, the study lacks a discussion on mmWave-specific FL applications and mmWave based concepts carried through in 6G networks. The study Boulogeorgos et al. (2021) explores the applications of THz wireless systems within the context of the Beyond fifth generation (B5G) era, delving into the emerging AI technologies that facilitate their implementation. However, the study does not cover the B5G network specifically, mmWave and FL enabled 6G networks. The authors in Pham et al. (2022) discuss the utilization of Mobile Edge Computing (MEC)-enhanced in Aerial Access Networks (AAN), commonly referred to as aerial computing, to explore the integration of FL in the context of 6G networks. However, the study does not discuss mmWave unitization in 6G networks. Similarly, the authors in Al-Quraan et al. (2023) seek to comprehensively explore cutting-edge FL applications within wireless technologies. Further, it aims to provide a comprehensive overview of the advancements and their significance. However, the areas related to mmWave spectrum and related concepts are only partially discussed. The study Duan et al. (2023) is one of the latest related work. It provides an in-depth examination of the latest technological advancements integrating FL and edge computing. Further, it covers a holistic perspective encompassing the intersection of FL and edge computing within the framework of 6G communication. However, the study does not discuss related issues related to mmWave communication in FL-enabled 6G networks. Likewise, the study Driss et al. (2023) explores the enhanced benefits of integrating FL across the entire protocol stack in the 6G technology.

Additionally, it highlights pivotal FL applications and delves into key concepts to offer valuable perspectives for future research and development endeavors. It contains only a brief discussion on mmWave utilization in FL enabled 6G network. The authors in Xiao et al. (2024) provide detailed analyses of various scenarios in FL over the air including, Multi-Input-Multi-Output (MIMO) and Reconfigurable Intelligent Surface (RIS). However, mmWave communication is only partially discussed with respect to RIS integration with FL. The study Lee et al. (2024) is a comprehensive survey on FL concepts in 5G and beyond network including network management, network core, network access etc. However, mmWave communication is discussed only partially in LTE communication domain. Notably, none of the previous reviews or surveys specifically cover the concepts involved in integrated scenarios of mmWave and FL enabled 6G communication.

## Research methodology
This methodology ensures a systematic approach to reviewing and analyzing literature on federated learning for millimeter-wave spectrum management in 6G networks, thereby

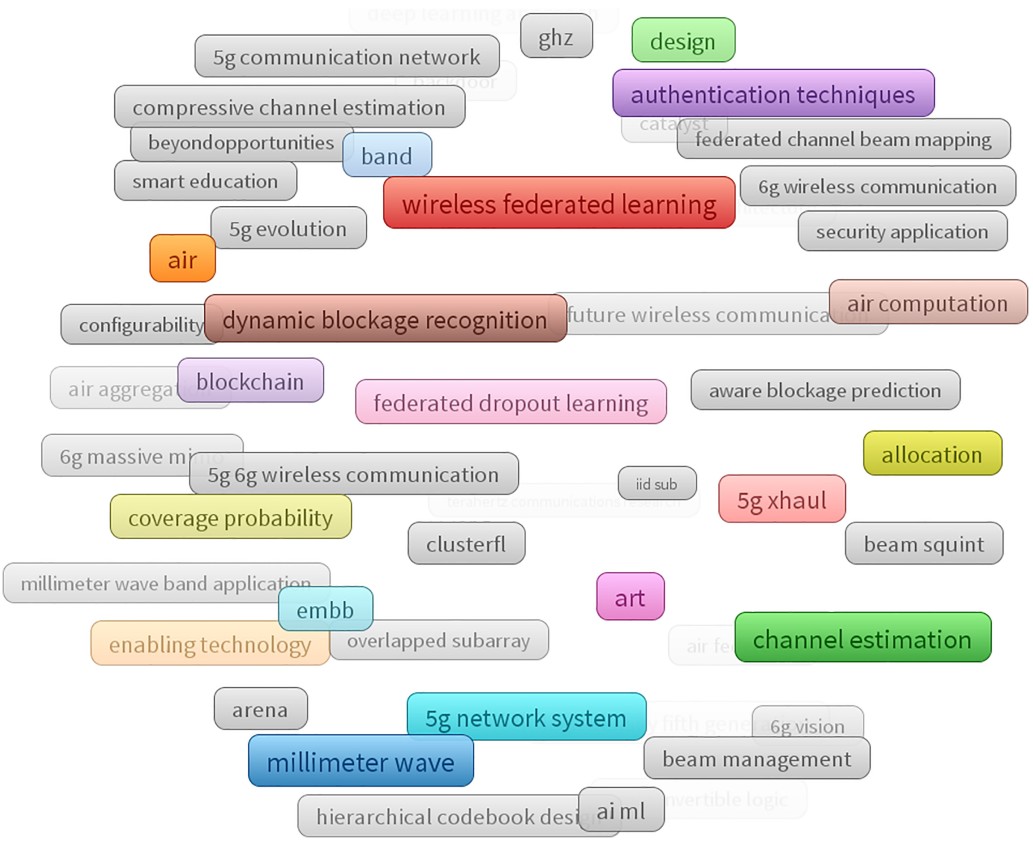

**Figure 1 VOSviewer based land scape of technologies covered in review.**

providing comprehensive insights into its applications, challenges, and future research directions. The following steps have been adapted for an exhaustive literature review:

- Online search: A systematic approach was adopted to conduct a review of the literature. Renowned academic databases, including SCOPUS, Web of Science (WoS), IEEE Xplore, ScienceDirect, MDPI, and Google Scholar, were searched using carefully chosen keywords such as "Federated Learning," "Millimeter-Wave Spectrum," "6G Networks," "FL Applications," "Challenges in FL," and "Open Research Issues in FL." The search time spanned from 2019 to 2024, ensuring the inclusion of recent advancements in the field.

- Article selection: In selecting relevant publications, thoroughly scrutinize titles, abstracts, and keywords to ascertain their alignment with FL in the context of millimeter-wave spectrum and 6G networks. Subsequently, exclusion criteria were applied meticulously to eliminate non-novel, duplicate, or irrelevant publications, thus refining the selection to encompass only the most pertinent contributions.

- Analysis: The study employed bibliometric analysis tools such as VOSviewer to comprehensively analyze articles' distribution based on titles and keywords, as shown in Fig. 1. By examining the distribution of relevant articles from various publishers,

the trends and patterns in the literature pertaining to FL for millimeter-wave spectrum in 6G networks were identified. Moreover, from the VOSviewer based landscape of technological terms, the keywords can be classified as;

- ○ Core technologies: Millimeter wave, 5G network, 5G/6G, enabling technology, wireless federated learning, federated dropout learning,
- ○ Network and spectrum: Dynamic blockage recognition, coverage probability, allocation, beam management, channel estimation, beam squirt, overlap subarray,
- ○ Privacy and security: authentication techniques, security applications,
- ○ Applications and use cases: smart education, art, design,
- ○ Emerging trends: 6G Massive MIMO, Blockchain, AI/ML.

- Synthesis: The literature review and analysis in this study reveal several findings regarding the applications, challenges, and open research issues associated with FL in leveraging millimeter-wave spectrum for 6G networks. Additionally, the study identifies challenges, including bandwidth consumption, power consumption, and synchronization requirements, while also providing a way forward for further investigation into areas like dynamic synchronization and bandwidth optimization for FL-enabled millimeter-wave communication in 6G networks. The overall research methodology opted in this research is depicted in Fig. 2.

## Organization and contributions

It is pertinent to highlight that almost all existing reviews and surveys have partially covered the topic. Moreover, the existing research lacks insight into the latest challenges and open research issues.

- *FL enabled 6G networks ("FL Enabled 6G Networks"):* This section initially discusses the basic concepts of FL; thereby, to establish FL potential, this section critically the FL application for 6G domains; including, enhanced Mobile Broad-Band (eMBB), Un-Conventional Data Communications (UCDC), Secure Ultra-Reliable Low-Latency Communications (SURLLC), Three-Dimensional Communications (3DCom) and Big Communications (BigCom).
- *Millimeter-Wave Applications in FL Enabled 6G ("mmWave Applications in Fl Enabled 6G"):* This section describes mmWave utilization for 6G networks including innovative concepts such as mmWave spectrum for integrated 6G, mmWave Chip for hybrid Massive MIMO (M-MIMO), sub-6GHz and mmWave dual-band antennas, joint venture of BF, mmWave and M-MIMO, wireless backhaul with mmWave and BF design and process.
- *Challenges ("Open Research Issues"):* This section highlights the primary challenges and limitations in embracing mmWave with FL; including bandwidth consumption, energy consumption and synchronization requirements.

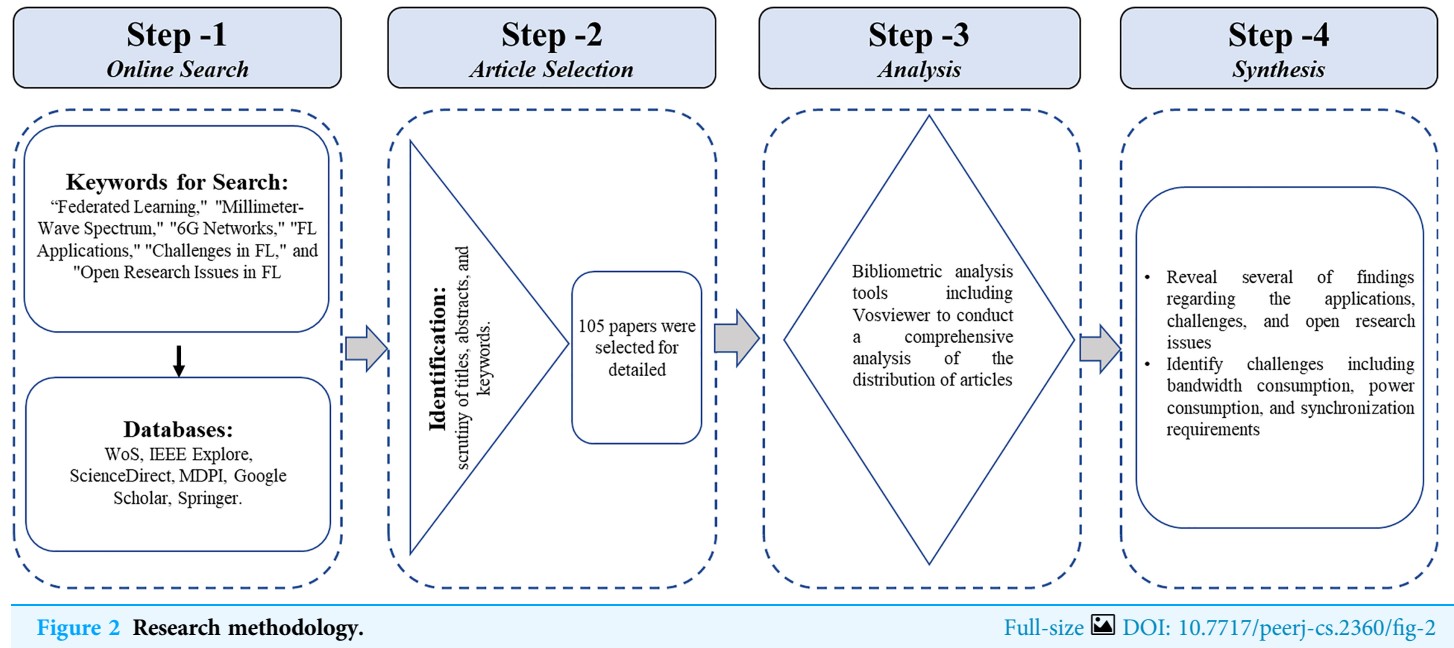

**Figure 2  Research methodology.**

- *Proposed way forward ("Conclusion"):* This section provides a way forward based of FL integration in mmWave communication architecture. The proposed approach is named Federated Energy-Aware Dynamic Synchronization with Bandwidth-Optimization (FEADSBO).

- *Future research directions ("Way Forward"):* Thereby, this review identifies the specific future research areas in FL enabled mmWave; including, FL for high-frequency mmWave communication, FL for energy efficiency in massive antenna systems, secure integration of mmWave in FL enabled 6G and mobility in mmWave with FL enabled 6G.

- *Conclusion ("Open Research Issues"):* The article concludes in this section with a brief on the aim of this review and the corresponding contribution to fill the research gap.

## FL ENABLED 6G NETWORKS

In 6G communication, DL and ML offer distinct advantages and disadvantages, making them suitable for network management and optimization aspects (*Noman et al., 2023*). ML, with its ability to process structured data and its relatively lower computational requirements, is well-suited for tasks like predictive maintenance, resource allocation, and anomaly detection in 6G networks. It is also more interpretable, allowing network engineers to easily understand and tweak models for specific tasks (*Tayyab et al., 2023*). However, the performance of ML can be limited in handling the vast and unstructured data expected in 6G systems, where more complex and nuanced patterns need to be identified (*Jawad, Maaloul & Chaari, 2023*). In contrast, DL excels in processing large-scale, high-dimensional, and unstructured data, such as images, video, and massive sensor

data streams (*Boahen et al., 2022*). Therefore, DL is considered as a potential technology for applications like intelligent traffic management, automated network slicing, and real-time service customization in 6G. DL models, such as neural networks, can automatically learn features from data, leading to higher accuracy in complex scenarios (*Ozpoyraz et al., 2022*).

However, DL comes at the cost of higher computational complexity, requiring more powerful hardware and potentially leading to higher latency, which may be a challenge for real-time 6G applications. In comparison to traditional ML and DL methods, which often require centralized data aggregation for training, while; FL offers a more scalable and privacy-preserving solution for 6G (*Zhang, Rahman & Qamar, 2023*). Therefore, FL is increasingly being proposed for 6G networks due to its unique approach to distributed machine learning, where data remains decentralized. This is particularly advantageous in 6G, which is expected to handle massive amounts of data from diverse and geographically distributed devices. The key advantage of FL in 6G networks is its ability to enhance privacy and security since the data never leaves the local devices, reducing the risk of data breaches and complying with stringent data protection regulations. Additionally, FL can reduce latency and bandwidth consumption by minimizing the need for data transfer to centralized servers (*Kazmi et al., 2024*). The foundational concept in FL is the distributed learning on localized datasets to develop local ML models and further aggregation of local models to formulate a Global model. Therefore, the overall data sent to the centralized server is substantially reduced; thus, the main attraction in FL is reduced network pressure associated with traditional AI implementations. Moreover, FL ensures end-point data protection by allowing sharing of only locally trained models (*Pandya et al., 2023*). A brief elaboration of FL can be visualized by assuming $N$ number computing devices $C_n \ni n = 1, \ldots, N$ with specific $D_n$ data volumes. The local model contains $w$ weights which are trained through inputs of data elements $x_i \ni i$ is the number of selected features. Thus, the fundamental FL averaging concept can be denoted as Eq. (1):

$$w^F = \sum_{n=1}^{N} \alpha_n . w_D^n \qquad (1)$$

where, $\alpha_n$ is the weighted average as given in Eq. (2).

$$\alpha_n \ni \left[ \sum_{n=1}^{N} \alpha_n \right] = 1. \qquad (2)$$

The aggregation process aims to keep the training loss minimum—spectrum's diverse range, progressing from radio waves to gamma rays. Initially, radio waves were the foundation, enabling long-range communication. As technology advanced, microwaves were employed for shorter distances, providing higher data rates—the utilization of infrared waves allowed for short-range communication, such as in remote controls. Recently, advancements in wireless communication have explored higher frequencies, including millimeter waves and even gamma rays, to meet the growing demand for data-intensive applications and achieve faster transmission rates. In view of unconventional and distributed FL architecture, this technology has become a core focus of

almost all AI-embraced scientific fields. Due to its distributed and heterogeneous nature, wireless mobile networks are a highly researched area for embracing FL technology (*Xu et al., 2023*).

In line with wireless communication, mobile wireless networks (MWN) have been standardized into generations from 1G to 6G. Likewise, this evolution has been marked by advancements in IEEE standards corresponding to radio frequency (RF) utilization. In the 1G era, standardized by IEEE 802.11, analog cellular networks operated at low frequencies, mainly in the 800 MHz range. The transition to 2G saw the introduction of digital communication, following GSM standards (*e.g.*, IEEE 802.16) and utilizing frequencies around 900 MHz. 3G networks, guided by standards like IEEE 802.20, embraced higher frequencies in the 2 GHz range for enhanced data rates and multimedia applications. 4G, governed by IEEE 802.16m and LTE standards, utilized a range of frequencies, including the 2.5 GHz band, enabling faster data transmission and improved network efficiency. The impending 5G era, defined by standards like IEEE 802.11ac, exploits higher frequencies, such as millimeter waves around 28 GHz, for ultra-fast data speeds and low latency. Anticipated 6G networks, still in conceptual stages, are expected to explore even higher frequencies, possibly into the terahertz range, to cater to emerging technologies and applications demanding unprecedented data rates and connectivity. Many research communities and standardization bodies are focusing on the forthcoming 6G mobile generation (*David & Berndt, 2018*; *Alsharif et al., 2020*; *Yang et al., 2019*). It asserted that a spectrum above mmWave frequency bands, precisely from 100 GHz up to 10 terahertz (THz) (*i.e.*, THz spectrum), is optimum to facilitate the next decade user's particulars and machine-type communication activities (*Kleine-Ostmann & Nagatsuma, 2011*). The large array of active antennas configured with THz spectrum and untangle signal processing techniques would benefit many upcoming unconventional applications (*Huang & Wang, 2011*). Remote presence, a holographic, and digital replica are prime examples. However, the notion is still in the early stage, and exploration in different areas, for example, scalability aspects, security, circuit architecture and complications, analysis of channel path loss expressions besides the administration of mobility management protocols are required (*Kemp et al., 2003*; *Ren et al., 2019*). 6G communication is the future of MWN, which is evolving with specific domains; including, enhanced Mobile Broad-Band (eMBB), Un-Conventional Data Communications (UCDC), Secure Ultra-Reliable Low-Latency Communications (SURLLC), Three-Dimensional Communications (3DCom) and Big Communications (BigCom) (*Kazmi et al., 2023a*).

## FL IN eMBB

eMBB, featuring advanced smartphones, dynamic gaming, and high-res multimedia, exhibits asymmetry in data rate needs, ranging from Mbps to 1+ Gbps. In the context of eMBB applications, FL supports efficient model training while minimizing the need to transmit large amounts of sensitive data over the network. FL enhances the adaptability of models to the dynamic nature of eMBB by allowing continuous learning from data generated across the network (*Guo et al., 2022*). The authors in *Hong, Park & Choi (2023)* introduce a novel over-the-air aggregation framework for FL in broadband wireless

networks. Unlike conventional FL setups, this framework accommodates local datasets on edge devices and base stations, thereby enhancing the efficiency and collaboration in model training across the network.

The integration of FL in eMBB aligns with the goals of achieving robust and secure communication in 6G networks while accommodating modern applications' diverse data rate requirements. Similarly, the study *Balakrishnan et al. (2020)* introduces resource management schemes that consider the importance of computing communication and data and optimizing training metrics. The proposed algorithms demonstrate a significant 4x–10x reduction in convergence time without compromising test performance, achieving a balance between model effectiveness and overall training efficiency on benchmark datasets.

## FL in SURLLC

In 6G networks, FL plays a pivotal role in SURLLC, which encompasses applications in smart tools, industries, and healthcare. FL facilitates collaborative model training across distributed devices, ensuring data privacy and security in SURLLC applications. The authors in *Khowaja et al. (2021)* proposed a Distributed Federated Learning (DBFL) framework that addresses challenges associated with long-range communication by mitigating the need for devices to increase transmission power, thereby alleviating energy efficiency concerns. This framework is designed to overcome connectivity issues for distant devices and seamlessly integrates with a mobile edge computing architecture. DBFL facilitates distributed communication among devices by utilizing clustering protocols, offering an efficient solution for collaborative learning while minimizing energy consumption. This decentralized approach allows SURLLC devices to collectively improve model accuracy without transmitting sensitive data to a centralized server, mitigating latency concerns. In smart tools and industries, FL enables real-time decision-making with low-latency communication, enhancing the reliability of critical processes. The study *Lu et al. (2020)* introduces the concept of Digital Twin Wireless Networks (DTWN), integrating digital twins into wireless networks to shift real-time data processing to the edge plane. This approach involves implementing a blockchain-empowered federated learning framework within DTWN, fostering collaborative computing to enhance system reliability, security, and data privacy. This innovative combination improves the overall functionality of the wireless network. In healthcare, FL supports secure collaborative learning on patient data distributed across devices, ensuring both reliability and privacy in the delivery of critical medical services within the 6G framework.

## FL in 3DCom

In the 3DCom of 6G communication, FL can potentially play a pivotal role in addressing challenges associated with airborne, multi-dimensional and high-rise platforms such as drones and underwater communication. FL enables decentralized machine learning models to be trained on these platforms, allowing them to adapt and learn from local data without transmitting sensitive information to a centralized server. This is particularly crucial for 3DCom due to its spatiotemporal characteristics, ensuring that machine

learning models can adapt dynamically to the changing environmental conditions of airborne and underwater scenarios. The study *Qu et al. (2007)* introduces an innovative concept known as AGIFL (Air-Ground Integrated FL), which seamlessly combines air-ground integrated networks with FL. Within the AGIFL framework, the adaptable and on-demand 3D deployment of aerial nodes, such as unmanned aerial vehicles (UAVs), enables all nodes to collectively train a proficient learning model through FL (*Sandamini et al., 2023*). Similarly, in order to overcome UWA tough conditions, the authors in *Zhao et al. (2021)* introduce C-DNN, a deep neural network-based receiver designed for underwater acoustic chirp communication. They also present an innovative approach by combining federated meta-learning with acoustic radio technology to enhance the performance and generalization of the DL model. The study provides tractable expressions for the convergence rate of FL in a wireless network by considering scheduling ratio, local epoch, and data volume on individual nodes. The potential outcome is improved, efficient, and reliable data transfer for underwater communication systems. However, challenges in this approach includes, potential complexities in coordinating and managing the federated meta-learning process, and limitations in addressing real-world variations in underwater environments. FL enhances privacy and security by keeping sensitive data local, mitigating potential risks associated with data transmission over these specialized 6G communication channels. Leveraging FL in 3DCom of 6G networks facilitates efficient and adaptive machine learning applications on diverse smart platforms.

## FL in UCDU

FL in the context of UCDC within 6G networks, offers an innovative approach to decentralized machine learning. UCDC, being an open-ended edge technology, benefits from the ability to train models across a distributed network of unconventional devices, such as smart human bond applications. This enables the dynamic and diverse nature of UCDC to be harnessed for collaborative learning without centralizing sensitive data. The authors in *Sozinov, Vlassov & Girdzijauskas (2018)* assess the effectiveness of FL in training a classifier for Human Activity Recognition (HAR), contrasting its performance with centralized learning. The findings indicate that while FL yields slightly lower accuracy than centralized models for human activity recognition, the difference is deemed acceptable. Likewise, the study *Ouyang et al. (2021)* introduces ClusterFL, a FL system designed for HAR applications. It utilizes a unique clustered multi-task federated learning framework to enhance model accuracy by efficiently capturing inherent clustering relationships among data from various nodes, minimizing communication overhead. The open-ended nature of UCDC aligns well with FL adaptability, facilitating secure and efficient attribute selection for authentication and confidentiality in this evolving technological landscape.

## FL in BigCom

FL in the context of BigCom in 6G contributes to a comprehensive global communication paradigm by decentralizing machine learning processes. This approach enables devices to collaboratively train models without transmitting raw data to a centralized server, ensuring privacy and security. The authors in *Chen, Xiao & Pang (2022)* advocate for implementing

FL in Low Earth Orbit (LEO)-based satellite communication networks to enhance support for highly interconnected devices with intelligent adaptive learning and mitigate costly traffic. Moreover, the study comprehensively reviews the current state-of-the-art LEO-based satellite communication and explores relevant ML techniques (*Siddiqui et al., 2022*). In 6G BigCom, FL accommodates the diverse aspects of communication technologies, allowing for dynamic and efficient learning across various devices and applications.

Similarly, integrating the Space-air-ground Integrated Network (SAGIN) with satellite, aerial, and terrestrial networks is enhanced through the innovative use of FL, a distributed learning method. This approach intelligently addresses resource scheduling challenges in SAGIN while ensuring security and safeguarding user privacy (*Tang et al., 2022*). The distributed nature of FL aligns with the need for robust communication in the face of evolving challenges, such as those presented by eMBB and other components of 6G. By fostering collaboration and learning across a globally distributed network, FL becomes an integral part of the holistic approach to communication in the BigCom paradigm of 6G.

## MMWAVE APPLICATIONS IN FL ENABLED 6G

The involvement of wireless information transfer among several IoT products in different verticals, for instance, transportation, supply chain, robotic activities, *etc.*, are expected to augment the data rates exponentially and need solutions that can tackle the extensive throughput stipulation and future extreme BW applications (*Akdeniz et al., 2014*; *Ibrahim & Hassan, 2019*). However, the existing 5G network design is promising for ultra-fast data rates and very low latency by exploiting the mmWave spectrum (*Ford et al., 2017*). mmWave refers to a specific range of radio frequencies between 30 and 300 gigahertz (GHz). These waves have much shorter wavelengths than those used in traditional wireless communication, which allows large amounts of data at much higher speeds. Moreover, mmWaves can be analogized as a highway with many lanes; because there are more lanes, more cars (or data) can travel simultaneously, leading to faster traffic flow (or data transmission). However, mmWaves have limitations in that it cannot travel as far or easily pass through obstacles like walls or trees, making them more suitable for dense urban environments (*Akyildiz, Han & Nie, 2018*). The key concepts related to mmWave include; M-MIMO, Backhaul, Dual-band Antennas, beamforming, *etc*. M-MIMO involves the use of a large number of antennas at the base station to serve multiple users simultaneously. Similarly, Backhaul refers to the communication link between the central network and the individual base stations. In the context of mmWave, high-capacity backhaul is essential to handle the massive data traffic generated by 5G and 6G networks. Likewise, Dual-band antennas are designed to operate on two different frequency bands, typically combining mmWave with a lower frequency band. This allows for both the high-speed benefits of mmWave and the broader coverage of lower frequencies. Beamforming is a technique where signals are directed precisely towards a specific user, rather than broadcasting in all directions. In mmWave, this focused transmission helps overcome signal blockages and improves data speeds (*Senger & Malik, 2022*).

In the 6G networks, mmWave technology is expected to be crucial in enabling ultra-fast, low-latency communications. This is particularly important for applications like

augmented reality, autonomous vehicles, and massive machine-type communications (like the Internet of Things), where speed and responsiveness are critical. However, mmWaves have a limited range, and implementing this technology on a large scale will require significant infrastructure changes, including installing many more small cells to ensure consistent coverage (*Xue et al., 2024*). Similarly, mmWave technology can be key in providing backward compatibility between 5G and 6G networks. While 6G aims to enhance the capabilities of mmWave by using higher frequency bands, mmWave can still operate within the frequency spectrum used by 5G, particularly in the lower end of the mmWave range (24–40 GHz). This overlap allows for seamless communication between 5G and 6G devices, enabling 6G networks to support legacy 5G devices and services. Additionally, the flexible architecture of mmWave technology allows it to adapt to various modulation and coding schemes used in 5G, ensuring that 6G networks can accommodate and optimize 5G transmissions without requiring a complete overhaul of the existing infrastructure (*Rajatheva et al., 2004*). In FL enabled 6G networks, mmWave technology is crucial in supporting the high-speed, low-latency communication required for distributed machine learning processes. mmWave operates in the 30–300 GHz frequency range, providing large bandwidths that enable ultra-fast data transfer, essential for transmitting the vast amounts of data involved in FL across devices. This high capacity and speed are vital for efficiently aggregating and processing data from multiple edge devices in real-time, allowing for more responsive and accurate AI models (*Rao et al., 2023*). In 6G, mmWave will also support applications like augmented reality (AR), virtual reality (VR), and smart cities, where FL can enable more personalized and privacy-preserving services. Additionally, FL in mmWave-enabled 6G networks can optimize network performance for heterogeneous networks by enabling real-time learning and adaptation to rapidly changing environments, such as those encountered in smart cities or autonomous vehicles (*Ji, Jia & Chen, 2019*). In subsequent sub-sections, we discuss the concept of mmWave in 6G communication in the context of FL.

## FL in mmWave spectrum for integrated 6G networks

The rapid-paced development of new technologies and smart modes of communication, as discussed in the 6G communication domains, is poised to overwhelm and strain the capabilities of the existing 5G network infrastructure, resulting in congestion and inadequate performance (*Hong et al., 2021*). The upcoming 6G will be a network of intelligent mobile communication, where almost everything will communicate with each other or the environment (*Giordani & Zorzi, 2020*). In this context, mmWave, and sub-mmWave technologies have the full potential to raise the bar and set new standards of intelligent communication. However, unconventional applications and heterogeneous architectures require highly dependable and distributed approaches to fully harness mmWave potential for 6G communication (*Qamar et al., 2018*).

AI, a revolutionizing concept, is a highly researched technology for integrating various technologies in 6G intelligent communication. FL is a privacy-protected and distributed approach that can provide an intelligent integration of mmWave with 6G communication. The authors in *Catak et al. (2022)* suggest an adversarial attack mitigation scheme ML

models for mmWave beam prediction in 6G. The method involves gradient sign techniques; however, this method requires further enhancement to avoid the trivial vulnerability of centralized DL/ML models. FL is a potential candidate to rescue the situation in combination with security mechanisms such as differential privacy (DP). It is a challenge to model the space-time non-stationarity properties of UAV based mmWave communication in 6G, including average fading time, envelope crossing rate, doppler power density and space-time frequency correlation (*Bai et al., 2021*). The authors in *Qi, Liu & Yang (2020)* utilize FL for dynamic adaptability to mobility patterns for improved hand-in mmWave communication. Moreover, the scheme is compatible with the client, who has limited storage capacity.

Similarly, the study *Salehi et al. (2022)* proposes a FL approach to expedite sector selection in the mmWave band for vehicular mobility by leveraging machine learning techniques that integrate data from LiDAR, GPS, and camera sensors. This includes a multimodal FL framework for collaborative model training among vehicles. Validation on a real-world dataset shows a significant 52.75% reduction in sector selection time compared to the 802.11ad standard, while maintaining 89.32% throughput with globally optimal solutions. However, this approach depends on the quality and availability of non-RF sensor data. Additionally, the effectiveness of the proposed solution is constrained by security issues inherent in sharing model weights among vehicles.

In mmWave systems, hybrid precoding architecture is considered suitable for mm-wave systems to achieve high beamforming gain with reduced hardware complexity, but channel estimation becomes challenging due to the split between analog and digital domains. The authors in *Zhao et al. (2022)* proposed a FL-based channel state information (CSI) estimation and feedback (FCEF) scheme, where users train local models and exchange parameters with the base station, reducing transmission overhead compared to centralized learning. However, with rapidly changing channels or highly dynamic user distributions, the effectiveness of the local model updates could be compromised, leading to degraded performance in CSI estimation and feedback. Similarly, the study in *Al-Abiad, Hassan & Hossain (2023)* introduces a resource-efficient FL framework tailored for millimeter-wave aerial-terrestrial integrated networks. It leverages decentralized model dissemination through UAV-to-UAV and device-to-device communication, enabling increased participation of user devices without relying on a central server. However, reliance on UAVs and complex scheduling algorithms can pose implementation challenges, particularly in real-world scenarios with dynamic network conditions and hardware constraints. The above analysis is summarized in Table 3.

### FL in mmWave chip for hybrid M-MIMO

The mmWave chip integrates both digital and analog components to efficiently process signals in M-MIMO systems, which are essential for next-generation wireless communication networks. By leveraging mmWave frequencies and hybrid beamforming techniques, this chip can support high data rates and enhance the capacity and coverage of wireless networks. A modification in the antenna design is also required to prosperously conduct the mmWave RF signals, especially sub-mmWave signals, in a closely packed

**Table 3 FL approaches in mmWave spectrum for integrated 6G networks.**

| Ref. | Approaches | Advantages | Limitations |
|---|---|---|---|
| Catak et al. (2022) | Gradient sign technique | • mmWave beam prediction in 6G<br>• Adversarial attack mitigation scheme ML models | • Does consider trivial vulnerabilities of ML<br>• Lack of transferability in gradient sign technique |
| Qi, Liu & Yang (2020) | Dynamic adaptability to mobility patterns with FL | • Improved hand-over<br>• Low storage utilization<br>• Improved QoS | • Scalability<br>• Security considerations related FL such as DoS attacks |
| Salehi et al. (2022) | Sector selection in mmWave band | • Significant reduction in sector selection time<br>• Least effect on throughput | • Dependent on non-RF sensor data<br>• Security issues inherent in sharing model parameters |
| Zhao et al. (2022) | CSI estimation and feedback | • Reduced transmission overhead compared to centralized learning | • Rapidly changing channels or highly dynamic user distributions can degrad performance |
| Al-Abiad, Hassan & Hossain (2023) | Millimeter-wave aerial-terrestrial integrated networks | • Increased participation of user devices without relying on a central server | • Implementation challenges due to dynamic network conditions and hardware constraints. |

hybrid M-MIMO system. ML algorithms can be employed for intelligent signal processing, adapting dynamically to varying channel conditions and optimizing beamforming parameters for improved system performance. Similarly, DL models can aid in complex signal processing tasks, such as channel estimation and interference mitigation, enabling the chip to achieve higher efficiency and reliability in M-MIMO systems. However, the centralized architecture of DL/ML can pose challenges related to processing and power consumption at the chip level. Therefore, FL is a potential distributed computing solution for resource constraints in the mmWave chips. The authors in *Elbir & Coleri (2020)* presented a FL based hybrid beamforming in mmWave. Here, the model training is performed at base stations (BS), where input data is the channel data used for analog beamforming. This technique aims to counter bandwidth consumption during data collection in traditional centralized learning ML/DL approaches.

The study *Onizawa et al. (2019)* suggests a training chip by utilizing a mainstream 65-nanometer CMOS technology for ML. The chip employs invertible logic and stochastic computing to achieve training without backpropagation. It enables direct extraction of weight values from input/output training data with low precision, making it well-suited for inference tasks. CMOS is developed for edge inference through a pre-trained model. However, the study established the effective utilization of CMOS-based processing in memory (PIM) Computing for model training and learning in FL applications (*Qamar et al., 2023*). The authors in *Vu et al. (2020)* introduce an innovative approach for cell-free massive multiple-input multiple-output (CFmMIMO) networks, specifically designed to facilitate any FL framework. The proposed scheme ensures the occurrence of individual iterations, rather than all, within an extensive coherence time to ensure the stable operation of the FL process.

**Table 4** FL approaches in mmWave chip for hybrid M-MIMO.

| Ref. | Approaches | Advantages | Limitations |
|---|---|---|---|
| *Elbir & Coleri (2020)* | Hybrid beamforming at base stations with channel data | • Reduced bandwidth consumptions <br> • Scalable as compared to ML/DL | • Vulnerable to adversarial attack <br> • Accuracy issues due to Non-IID data |
| *Onizawa et al. (2019)* | 65-nm CMOS technology for ML/FL | • Establishes the effective utilization of CMOS-based PIM in FL applications | • Vulnerable to inference attacks <br> • Security implementation such as DP can increase complexity |
| *Vu et al. (2020)* | CFmMIMO | • Individual iterations within a coherence time <br> • Joint optimization of FL and transmit power | • Sensitive to network dynamics <br> • Complexity in model generalization |
| *Farooq et al. (2023)* | HD and FD communication schemes | • Enables privacy preservation and communication efficiency | • Self-interference or small FL model updates can affect the performance of FD |
| *Zhong, Yang & Yuan (2022)* | Over-the-air federated multi-task learning | • Leverages analog superposition for computation | • Dealing with a large number of devices can potentially impacting the effectiveness |

Similarly, the authors in *Farooq et al. (2023)* proposed half-duplex (HD) and full-duplex (FD) communication schemes to handle simultaneous FL and non-FL user groups sharing the same resource, enabling privacy preservation and communication efficiency in MIMO system. Meanwhile, self-interference or small FL model updates can affect the performance of full-duplex communication. Moreover, the generality and scalability of this approach require the evaluation of real-world heterogeneous scenarios.

Similarly, the study *Zhong, Yang & Yuan (2022)* discusses Over-the-Air Federated Multi-Task Learning (OA-FMTL) over the MIMO multiple access channel, leveraging analog superposition of electromagnetic waves for computation. It introduces a novel model aggregation technique to align local gradients from different devices, mitigating the straggler problem arising from channel heterogeneity. However, dealing with a large number of devices can potentially impact the effectiveness of the proposed approach. The above discussion is summarized in Table 4.

## Sub-6 GHZ and mmWave dual-band antennas

The NR 5G mobile communication is compatible with earlier technologies, and simultaneous operation of both sub-6 GHz and mmWave spectrums on the same antenna nodes is desirable. Indeed, new shared aperture antennas that support the operation of both frequency bands are a serious challenge and have recently emerged as a hot topic (*Hasan, Bashir & Chu, 2019*). The major design issue is the dimension requirements, as the gap between both frequency bands is very large. Research in the sub-6 GHz spectrum for dual-band antennas typically focuses on the frequency bands, including low bandwidth (Sub-1 GHz for LTE) and mid bandwidth (1–6 GHz for WiFi). While dual-band antennas

are typically associated with sub-6 GHz frequencies, mmWave frequencies (24 GHz and above) are used for high-capacity and ultra-fast communication (*Alieldin et al., 2018*; *Sang et al., 2023*).

Utilizing sub-6 GHz channels for predicting millimeter wave (mmWave) beams and blockages holds promise for enhancing mobility and reliability in scalable mmWave systems. By employing a sufficiently large neural network, achieving high success probabilities in predicting mmWave beams and blockages is feasible, approaching near certainty (*Alrabeiah & Alkhateeb, 2020*). Here, the decentralized nature of FL can facilitate the aggregation of insights from multiple devices, leading to improved success probabilities in predicting mmWave beams and blockages, with the potential to achieve high accuracy and lower bandwidth consumption. The authors in *Al-Quraan et al. (2023)* introduce Radar-aided Dynamic Blockage Recognition (RaDaR), which combines radar data and FL to train a dual-output neural network model. This model can predict both blockage status and time concurrently for proactive handover or beam switching. This approach mitigates latency in 5G new radio procedures, ensuring a high-quality of experience for users.

Similarly, the authors *Chafaa et al. (2021)* introduced an FL framework for a wireless network with multiple communication links (access points and users). This approach involves individual access points training their local deep neural networks using local data, sharing only model parameters to achieve a collectively improved global model for predicting downlink mmWave beamforming vectors based on uplink sub-6GHz channels, thereby; enhancing data rate predictions.

However, due to the heterogeneous and varying nature of wireless network, the data is considered highly Non-Independently Distributed (Non-IID), which can severely affect FL's performance. The research *Bai et al. (2021)* proposes two personalized FL approaches to address non-IID effects. The first involves fine-tuning FL models on individual private datasets of clients, while the second employs Adaptive Expert Models for FL to directly predict the optimal mmWave beamforming vector from non-IID sub-6 GHz channel vectors generated by Deep-MIMO.

The study *Hu et al. (2023)* proposes combining multi-band- reconfigurable holographic surfaces (RHSs) and FL to offer precise and environment-adaptive user positioning services. It employs an FL framework for collaborative training of a position estimator, leveraging transfer learning to address the absence of position labels among users, and introduces a scheduling algorithm for base stations to select users for training, considering both FL convergence and efficiency. However, implementing such a system across diverse network environments can be challenging. Similarly, the study *Chafaa et al. (2022)* presents a self-supervised DL method for predicting beamforming vectors in mmWave communication, utilizing sub-6 GHz channels and DeepMIMO dataset. The extension to multiple links using FL efficiently predicts mmWave beams with limited local data. However, potential limitations may arise from network synchronization challenges and the need for extensive computational resources for training and coordination. Here, the potential challenge is the synchronization of multiple access points in FL setups. The above analysis is summarized in Table 5.

**Table 5  FL approaches in sub-6 GHZ and mmWave dual-band antennas.**

| Ref. | Approaches | Advantages | Limitations |
|---|---|---|---|
| *Al-Quraan et al. (2023)* | Dynamic blockage recognition | • Predict both blockage status and time concurrently<br>• Provide proactive handover and beam-switching | • Limited generalization in a dynamic environment<br>• Scalability |
| *Chafaa et al. (2021)* | Predict the downlink mmWave beamforming vectors from the uplink | • Exploit the available knowledge at sub-6 GHz<br>• Outperform centralized learning | • Dependency on Sub-6 GHz<br>• Sensitivity to local data quality |
| *Cheng (2022)* | Fine-tuning and Adaptive Expert Models | • Predicts optimal mmWave beamforming vector | • The approach is specific to MIMO-based systems |
| *Hu et al. (2023)* | Combine utilization of multi-band- RHSs and FL | • Offers precise and environment-adaptive user positioning services | • High complexity in diverse network environments |
| *Chafaa et al. (2022)* | Self-supervised DL | • Accurately predicts beamforming vectors in mmWave communication | • Synchronization issue in multiple access points in FL setups |

## Joint venture of BF, mmWAVE, and M-MIMO

Since the conventional sub-6 GHz spectrum is fully occupied, the researcher's community is performing extensive investigations on successfully utilizing higher frequency spectrum in the mmWave and mmWave operable frequency spots. Herein, both high-frequency radio wave signals are regarded as commendatory technologies for forthcoming mobile communication (*Song, Yang & Sun, 2017*). However, it is undesirable for the cellular data transmission environment due to its limited coverage support in a given geographical area (*Goudarzi et al., 2020*). Therefore, some advanced multi-beam or beam-steerable antenna schemes have been lately adopted to conduct upper spectrum transmissions with M-MIMO antenna in B5G cellular networks. The remote channel state information (CSI) inference involves DL structures in a communication system, to accurately assess channel conditions. DL/ML approach enables a more sophisticated and adaptable method for inferring CSI in remote communication scenarios where conventional models may fall short. The study in *Jiang et al. (2019)* aims to assess the effectiveness of a DL-based approaches in the context of mmWave multicellular networks. The study focuses on optimizing beamforming configurations through two neural networks trained to minimize mean square error (MSE). The first network considers requested spectral efficiency (SE) per active sector as input, while the second network addresses the corresponding energy efficiency (EE), enabling adaptive beamforming to account for channel and power variations. Nonetheless, the proposed approach achieve satisfactory performance in enhanced data rates at the cost of increased radiating nodes and blocking probability.

Similarly, utilizing FL has emerged as a promising approach to enhance the performance of learning-based mmWave BF systems for efficient link configuration. In the study *Elbir, Coleri & Mishra (2021)*, two distinct frameworks, namely model-based and

**Table 6 FL approaches in joint venture of BF, mmWave, and M-MIMO.**

| Ref. | Approaches | Advantages | Limitations |
|---|---|---|---|
| Jiang et al. (2019) | NN based adaptive beamforming to account for channel and power variations | • Achieves satisfactory performance in enhanced data rates | • Increased blocking probability can reduce the accuracy of NN model |
| Elbir, Coleri & Mishra (2021) | Multi-user Spatial Path Index Modulation and FL with dropout technique | • Users estimate beamformers by inputting their channel data<br>• Facilitating collaborative beamforming | • Limited consideration of RF chain<br>• Performance issues due to Non-IID data |
| Zhang et al. (2022) | Trigger attack with obstacle in certain locations | • Explores backdoor attacks, defenses<br>• Propose a new backdoor attack defense | • Complexity of real-world scenarios<br>• Generalization issues |
| Huang et al. (2022) | Joint device scheduling and receive beamforming | • Minimizes the FL convergence gap for devices under latency and power constraints | • Receive beamforming requires significant computational resources |
| Asaad et al. (2024) | OTA-FL | • Joint antenna selection and beamforming for model aggregation | • High complexity with the increase in the number of antennas and RF-chains |

model-free, are introduced for the design of beamformers in multi-user Spatial Path Index Modulation (SPIM) for joint mmWave-MIMO systems. The study employs FL coupled with dropout learning techniques. This approach involves training a learning model on the local datasets of individual users. These users estimate beamformers by inputting their channel data into the model, thereby; facilitating collaborative learning across the network.

Despite the potential benefits of FL, FL systems are susceptible to backdoor attacks during training in FL-based beam selection for mmWave systems. A backdoor attacker aims to inaccurate the out of ML model a predefined set of inputs. This manipulation is designed to undermine the model's performance on targeted sub-tasks while appearing normal for other inputs. The authors in Zhang et al. (2022) presented a backdoor attack strategy where the model activates upon encountering specific obstacles in designated locations. When the model processes input featuring these obstacles, the backdoor is triggered, causing the model to produce the predefined output beam specified by the attacker.

Similarly, the study Huang et al. (2022) proposed an approach that integrates joint device scheduling and receive beamforming to minimize the FL convergence gap over shared wireless MIMO networks, maximizing the number of weighted devices under latency and power constraints. However, the complexity of jointly optimizing device scheduling and receive beamforming requires significant computational resources.

Likewise, the study Asaad et al. (2024) investigates Over-the-Air FL (OTA-FL) in massive MIMO systems with limited RF-chains, addressing the challenge of joint antenna selection and beamforming for model aggregation. This scheme is based on a two-tier approach utilizing penalty dual decomposition and treating antenna selection as a sparse recovery problem using the Lasso algorithm. Here, scalability can become complex with

the increase in the number of antennas and RF-chains. The above discussion is summarized in Table 6.

## Wireless backhaul with mmWAVE

Currently, conventional backhaul wireless links are largely based on microwave frequency bands (often owned by operators) and fiber/copper cables (often leased) with varying proportions per operator and country area (*Allen, Chevalier & Bora, 2014*). Many other types of research and simulation tests have also been developed to evaluate small-cell wireless backhaul communication, including software-defined networking (SDN) based techniques (*Niephaus et al., 2015*; *Zhang et al., 2015*). In contrast, some recent research works using AI and ML on the agility of dynamic configuration of backhaul streams have also been discussed. Many mmWave-based SDN wireless backhaul access systems focus on optimizing different parameters, which are exhaustively discussed in the literature (*Santos et al., 2019*; *Santos, 2020*; *Camps-Mur et al., 2019*).

With the emergence of heterogeneously connected devices and advanced applications like IoT, V2V communications, and wearables, industry experts foresee complex management requirements in backhaul communication. FL resolves these privacy and bottleneck issues by training a global model and also plays a crucial role in optimizing wireless backhaul communication networks (*Chehri et al., 2023*). The authors in *Yang, Hong & Park (2021)* introduce an adaptive power allocation method, where each client dynamically allocates its transmit power based on the magnitude of the FL based gradient information, optimizing communication efficiency. The article *Wang et al. (2021)* explores the optimization of energy and time consumption in mobile-edge computing-enabled balloon networks, where high-altitude balloons serve as flying wireless base stations. The proposed solution employs a support vector machine (SVM)-based FL algorithm to proactively determine user associations, allowing high-altitude balloons to dynamically adjust resource allocation for computational tasks without transmitting historical associations or tasks.

The study *Feng & Mao (2019)* proposes a solution to the challenge of backhaul resource allocation in mmWave systems using deep reinforcement learning (DRL). By leveraging DRL, the system learns the blockage patterns and dynamics, enabling efficient allocation of backhaul resources to users with highly varying data rates. DRL models in real-world mmWave face potential difficulty in generalizing learned policies across different network scenarios; here, personalized FL approaches can be used to overcome the challenges related to generalization.

The study *Mahmood et al. (2024)* proposes (0.1 to 10 THz) Terahertz-based networks to improve FL convergence time and reduce training loss, particularly in 1 km links, by exploiting efficient data transmission conditions in comparison. This approach involves implementing FL with THz-based wireless backhaul and a virtual private network (VPN) infrastructure, catering to end-users for enhanced privacy and network efficiency. However, signal attenuation and hardware complexities can hinder the adoption of FL deployment in real-world scenarios. Similarly, the authors in *Khan et al. (2023)* propose a latency-aware vision-aided FL approach for predicting beam blockage in 6G wireless

**Table 7 FL approaches in wireless backhaul with mmWave.**

| Ref. | Approaches | Advantages | Limitations |
|---|---|---|---|
| *Yang, Hong & Park (2021)* | Binary gradient updating strategy in FL | • Optimizing communication efficiency<br>• Adaptive power allocation using CNN | • Sensitivity to channel conditions<br>• Computational overhead |
| *Wang et al. (2021)* | High-altitude balloons with SVM based FL | • Minimizes energy and compute time<br>• Dynamically adjusts resource allocation | • Does not consider non-IID data<br>• Lack of security consideration |
| *Feng & Mao (2019)* | Backhaul resource allocation using DL | • Learns the blockage patterns and dynamics for varying data rates | • Difficulty in generalizing learned policies across different network scenarios |
| *Mahmood et al. (2024)* | Terahertz-based networks | • Efficient data transmission conditions | • Hardware complexities can hinder the adoption in real-world scenarios |
| *Khan et al. (2023)* | Latency-aware vision-aided FL | • Mitigates, bandwidth inefficiency, and latency issues | • Increased complexity during coordination and synchronization |

networks. It utilizes multi-sensor data and advanced deep learning techniques by employing distributed learning on edge nodes for data processing and model training. This framework aims to mitigate communication costs, bandwidth inefficiency, and latency issues associated with centralized training; however, this approach can increase the complexity of coordination and synchronization across numerous edge nodes. The above analysis is summarized in Table 7.

## BF design and process

In multi-stream multi-user mmWave transmission, the BS with large-scale antenna elements simultaneously serves many users with the directed beam patterns (*Sun, Qi & Li, 2019*; *Kazmi et al., 2022*). A hybrid precoding mechanism is commonly adopted for mmWave M-MIMO communication, wherein analog precoding/BF generates directional beams under constant amplitude, limited phase shifter resolution constraints, and digital precoding multiplexed independent user data streams. Many M-MIMO hybrid precoding solutions have been proposed for mmWave wireless communication (*Alkhateeb et al., 2014*; *Xiao et al., 2016*; *Xiao, Xia & Xia, 2017*). One of the challenging aspects of BF is Beam squinting. This is a phenomenon associated with non-uniformities in the array elements, causing the beam to deviate from its intended direction. In 6G communication systems, beam training is crucial for optimizing the directional transmission and reception of signals between base stations and user devices. This involves dynamically adjusting the beamforming parameters, such as beam direction and width, to adapt to changing environmental conditions and mitigate beam squinting effects through adaptive algorithms and sophisticated signal processing (*Chen, Chen & Jiang, 2021*). Designing BF for large-scale antenna arrays, particularly with constraints such as limited radio frequency chains and the use of phase-shifter-based analog BF architecture, poses a significant challenge in millimeter-wave communication systems. Therefore, the complexity further intensifies in the presence of imperfect CSI. The study *Niephaus et al. (2015)* introduces a

**Table 8  FL approaches in mmWave BF design and process.**

| Ref. | Approaches | Advantages | Limitations |
| --- | --- | --- | --- |
| *Lin & Zhu (2019)* | DL for BF design | • Optimize the beamformer by considering hardware limitations and imperfect CSI | • Privacy issues due to conventional design |
| *Kim, Swindlehurst & Park (2023)* | BF vectors and the selection of devices | • Maximizes the count of selected devices<br>• Remain consistent with predefined MSE | • Imperfect channel state information<br>• Scalability |
| *Xiao et al. (2007)* | ISPW-ADMM | • Resilience in time-varying dynamic networks<br>• Rapid convergence | • Computational complexity<br>• Issues for inexact stochastic |
| *Elbir & Coleri (2021)* | Channel estimation in RIS assisted massive MIMO systems | • Significantly reduced overhead compared to conventional learning, | • Coordination in the FL process can impact the overall efficiency |
| *Jeon et al. (2020)* | Compressive sensing | • Accurate reconstruction of local gradient vectors | • Assumptions of sparsity and LMMSE can cause inaccuracies |

novel approach using DL for BF design to address this. The proposed method involves the development of an NN that can be trained to optimize the beamformer, considering hardware limitations and dealing with the challenges posed by imperfect CSI, ultimately maximizing the spectral efficiency. The effectiveness of FL aggregation is directly linked to the increased participation of devices. The article *Zhang et al. (2015)* presents a design based on BF vectors and the selection of devices for FL using over-the-air computation (AirComp). The results show that the approach maximizes the count of selected devices while adhering to a predefined target aggregation MSE.

Similarly, the authors in *Xiao et al. (2007)* introduce a fully decentralized FL framework featuring an inexact stochastic parallel random walk alternating direction method of multipliers (ISPW-ADMM). The findings indicate that this framework exhibits significant resilience against the effects of time-varying dynamic networks and stochastic data collection while maintaining rapid convergence. Leveraging the advantages of stochastic gradients and biased first-order moment estimation, the proposed framework is a dynamic approach for decentralized FL tasks across time-varying graphs.

Similarly, the authors in *Elbir & Coleri (2021)* introduce a FL framework for channel estimation in both conventional and RIS-assisted massive MIMO systems. They utilize a CNN trained on local datasets of users, avoiding the need to transmit them to the BS. Their approach demonstrates significantly reduced overhead compared to conventional learning, approximately 16 times lower, while maintaining performance close to CL. However, coordination in the FL process across numerous distributed users can impact the overall efficiency and scalability of the system.

Likewise, the study *Jeon et al. (2020)* introduces a compressive sensing approach for FL in MIMO communication systems. It addresses the challenge of accurately reconstructing local gradient vectors sent from wireless devices by establishing a transmission strategy for constructing sparse transmitted signals and proposing a compressive sensing algorithm for

the central server to iteratively find the linear minimum-mean-square-error (LMMSE) estimate of the transmitted signal. This approach relies on sparsity assumptions and linear minimum-mean-square-error estimation, which can be challenging in heterogeneous 6G networks. The above discussion is summarized in Table 8.

## CHALLENGES

In this section, this review highlights explicitly the challenges for the integration of mmWave technology in FL enabled 6G communication as follows:

### Bandwidth consumption by FL

The integration of mmWave technology into FL-enabled 6G communication is linked with substantial bandwidth consumption due to the continuous exchange of training details among edge devices. The inherent high data rates of mmWave communication cause bandwidth limitations, specifically during the phases of FL processes and updating (*Al-Quraan et al., 2023*). Therefore, the utilization of mmWave frequencies is dependent on the efficient data transfer to accommodate model updates exchanged among federated nodes. This increased demand poses a severe challenge to the seamless integration of mmWave into FL enabled 6G communication. Similarly, to address this, innovative solutions must be devised to optimize data transfer protocols (*Hafi et al., 2023*). Techniques such as advanced compression algorithms, adaptive modulation schemes, and prioritized data transmission strategies can potentially effectively manage the critical data volumes exchanged during FL operations over mmWave frequencies (*Li et al., 2023*). Additionally, integrating intelligent bandwidth management mechanisms and incorporating dynamic allocation and spectrum-sharing strategies are essential to mitigate potential congestion issues arising from the augmented bandwidth demand. Therefore, bandwidth limitation is a multifaceted challenge that requires advanced compression techniques with dynamic bandwidth management to achieve efficient and congestion-free FL processes in the context of mmWave-enabled 6G communication.

### Energy consumption by FL

Integrating mmWave technology into 6G communication, particularly in the context of FL, presents a significant challenge due to heightened energy consumption during training and data processing at the edge nodes. The resource-intensive nature of FL processing is compounded by the energy-demanding characteristics inherent in mmWave transmission, involving; DL based training at local devices and further, iterative model updates across distributed nodes (*Nguyen et al., 2021*). In mmWave communication, the shorter wavelengths and higher frequencies lead to increased path loss and susceptibility to atmospheric absorption, necessitating higher transmit power to maintain communication reliability. This inherent limitation of mmWave contributes to substantial energy requirements in the overall system (*Lim et al., 2020*). To address this challenge, the development of energy-efficient algorithms and protocols are required alongside standard FL processing over mmWave frequencies. One approach involves optimizing the model update process to minimize redundant transmissions, employing techniques such as

sparsity-aware algorithms to reduce the amount of data exchanged (*Dinh et al., 2020*). Additionally, incorporating adaptive power control mechanisms can help optimize the transmit power levels, dynamically adjusting them based on channel conditions to balance reliability and energy efficiency. Furthermore, the design of compression algorithms tailored for mmWave FL can significantly alleviate energy consumption. Thus, addressing the challenge of increased energy consumption in 6G communication with mmWave-enabled FL demands a holistic approach, from optimizing model updates to implementing adaptive power control to ensure sustainable and optimized energy usage in this complex and dynamic communication paradigm (*Liu & Simeone, 2020*).

## Sychnozation requirements in FL

The joint integration of mmWave technology and FL in 6G communication introduces the challenge of maintaining synchronization in communication and FL processing (*Rodríguez-Fernández, 2021*). Due to the significant susceptibility of mmWave signals to propagation conditions, special attention must be given to maintaining synchronization among federated nodes. This synchronization is paramount for the seamless and coherent aggregation of models distributed across various nodes in the FL processing. Similarly, ensuring a reliable synchronization in Heterogeneous FL enabled 6G networks is critical for achieving coherent collaboration among distributed nodes (*Qi et al., 2023*). Similarly, including satellite links in the BigCom domain of 6G networks introduces additional latency considerations, necessitating sophisticated synchronization protocols to account for varying transmission delays (*Giordani & Zorzi, 2020*). Therefore, the solution in this domain must exhibit high resilience and adaptability to ensure precise and timely coordination among distributed nodes throughout the FL process. Achieving synchronization is challenging in scenarios where propagation delays and signal distortions have high recurrent probabilities (*Chukhno et al., 2023*). The intricacies of mmWave communication, characterized by short wavelengths and susceptibility to blockages, further complicate the synchronization challenge. Consequently, the successful resolution of synchronization challenges assumes pivotal importance in unlocking the full potential of mmWave in FL enabled 6G communication.

## Security and privacy in FL

The integration of federated learning (FL) with millimeter-Wave (mmWave) technology in cellular networks, spanning 4G, 5G, and emerging 6G systems, presents significant security and privacy challenges (*Liu et al., 2020*). FL, which enables decentralized model training across devices without sharing raw data, is attractive for maintaining user privacy. The study in *Yang et al. (2022)* presents an estimation algorithm to prevent privacy leakage in Cybertwin-Driven 6G system. This infers local data distribution from clients without accessing their raw data. This approach addresses two scenarios in FL: one where the server receives individual trained models from each device, and another where it receives an aggregated model, with device selection strategies formulated to optimize training performance in both cases. However, the decentralized nature of FL introduces vulnerabilities, such as adversarial attacks where malicious participants can corrupt the

global model or infer sensitive information through model updates. The study in *Bárcena et al. (2023)* introduces a novel FL-as-a-Service framework designed for B5G/6G networks, particularly in a vehicular networking scenario by leveraging FL with eXplainable AI (XAI) models. This framework enhances both the accuracy of QoE predictions in local learning and the trustworthiness. Similarly, FL with mmWave technology, with its high-frequency bands, enhances network performance but also poses unique security risks, including increased susceptibility to eavesdropping due to the narrow beam and short-range characteristics of mmWave signals (*Catak, Catak & Moldsvor, 2021*). Additionally, the reliance on dense infrastructure for mmWave deployment raises concerns about physical security and the potential for infrastructure attacks. As cellular networks evolve from 4G to 6G, ensuring robust security and privacy in the integration of FL with mmWave requires advanced cryptographic techniques, secure aggregation methods, and enhanced physical layer security to mitigate these risks effectively.

# WAY FORWARD

Given the challenges and insights in this review, we suggest a way forward: Federated Energy-Aware Dynamic Synchronization with Bandwidth-Optimization (FEADSBO), that combines aspects of bandwidth optimization, energy efficiency, and synchronization in FL. Frequency synchronization ensures that the carrier frequencies of transmitters and receivers are aligned to enable successful signal reception. Precise frequency synchronization is essential in mmWave communication, where small frequency offsets can lead to significant performance degradation. Techniques such as carrier aggregation and frequency tracking algorithms achieve frequency synchronization in mmWave systems. The core functionality in FEADSBO is the dynamic adjustment of synchronization frequencies and bandwidth consumption based on energy and resource availability of participating devices in FL.

Similarly, FEADSBO optimizes bandwidth consumption by prioritizing the transmission of compressed updates, while adaptive synchronization thresholds dynamically adjust synchronization frequency based on network conditions and convergence speed. This comprehensive approach minimizes unnecessary communication overhead, conserves energy, and ensures timely updates, making FEADSBOa practical solution for FL in resource-constrained environments such as IoT, edge computing, and mobile devices. The proposed approach consists of the following components.

## Dynamic synchronization

In FEADSBO, participating devices dynamically regulate their synchronization frequency based on energy availability and computational resources, ensuring that energy-constrained devices synchronize less frequently to conserve power. Devices dynamically adjust the frequency of synchronization based on their energy availability and computational resources. Devices with sufficient energy and computational resources participate in more frequent synchronization rounds. The system can optimize bandwidth usage and conserve energy more effectively by allowing participating devices to adjust their synchronization frequency according to their energy constraints and computational

capabilities. This dynamic adjustment mechanism helps strike a balance between synchronization requirements and resource constraints, making mmWave-enabled FL more sustainable and efficient.

## Energy-aware bandwidth optimization

FEADSBO utilizes quantization and pruning-based model compression techniques to optimize bandwidth consumption during model updates. Prioritizes transmission of compressed model updates and aggregates them at the server side, reducing the overall data transmitted over the network. Compressed model updates reduce bandwidth consumption, allowing for efficient utilization of network resources. Quantization involves reducing the precision of numerical values in the model, typically from floating-point to fixed-point representation. This reduces the memory and computational requirements during model inference, making it more suitable for resource-constrained devices in federated settings. Likewise, pruning involves removing unnecessary parameters or connections from the model, effectively reducing its complexity and size. Pruning techniques can be applied globally or locally across layers, identifying and eliminating redundant information while preserving accuracy. Together, quantization and pruning-based compression techniques enable FL to achieve better scalability, faster inference times, and reduced communication overhead.

## Adaptive synchronization thresholds

FEADSBO dynamically sets synchronization thresholds based on network conditions and model convergence speed. Network conditions encompass several properties that can influence the synchronization thresholds in a dynamic system. These properties include network latency, bandwidth availability, packet loss rate, and network topology. Latency refers to the time delay between data transmission and reception and can vary based on the distance between devices, network congestion, and routing efficiency. Bandwidth availability determines the amount of data that can be transferred within a given timeframe, affecting the speed and efficiency of synchronization processes. Packet loss rate measures the percentage of data packets lost during transmission, impacting data integrity and reliability. Network topology defines the structure and connectivity of devices in the network, influencing communication paths and potential bottlenecks. By considering these properties, dynamic systems can adaptively adjust synchronization thresholds to optimize data exchange, minimize latency, mitigate packet loss, and enhance overall performance based on prevailing network conditions and model convergence speed in FL environments. Devices with faster convergence or stable network conditions can trigger synchronization less frequently, while devices experiencing slower convergence or fluctuating network conditions synchronize more frequently to ensure timely updates. Dynamic adjustment of synchronization frequency ensures timely updates while adapting to varying network conditions and device capabilities.

By integrating these above-discussed three aspects into the FEADSBO, FL systems can achieve improved efficiency in terms of bandwidth consumption, energy consumption, and synchronization requirements for mmWave spectrum utilization in 6G environment.

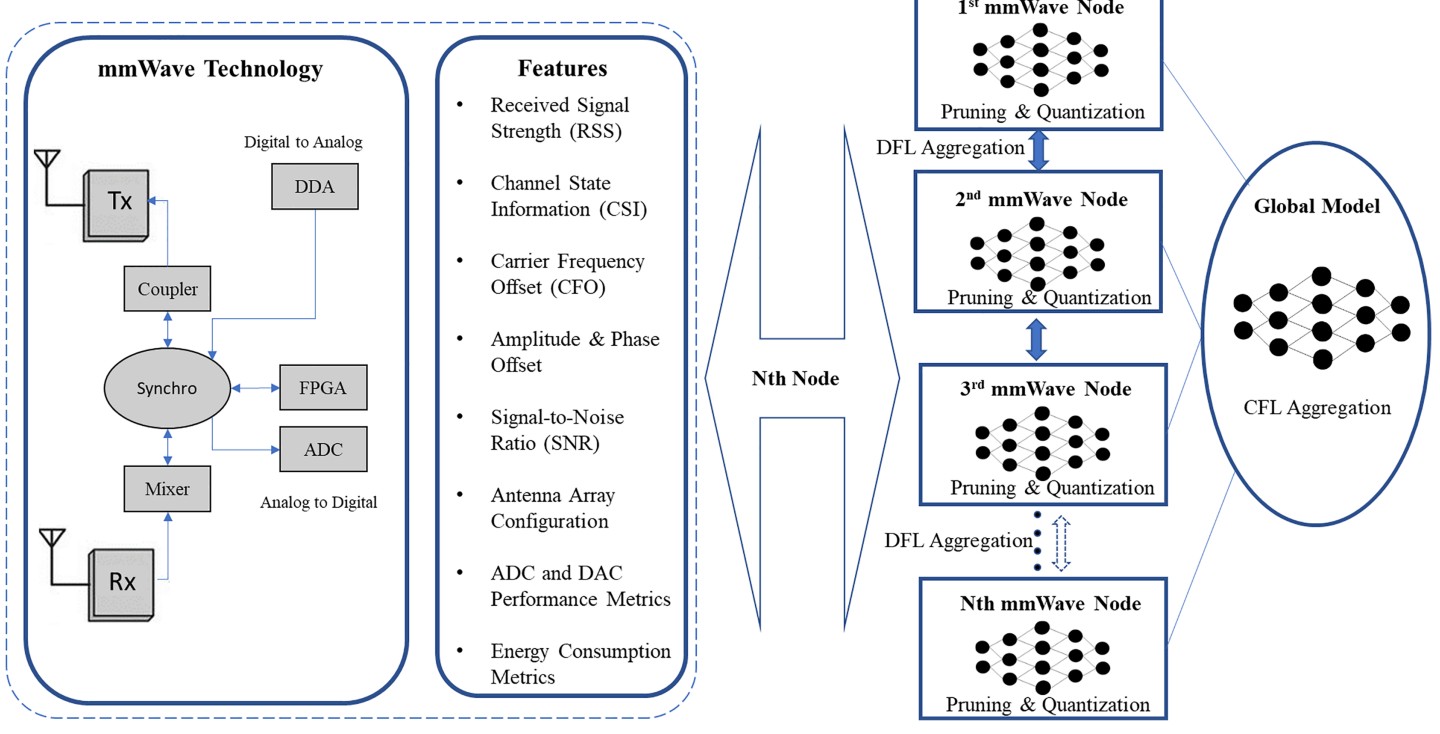

**Figure 3** Organization of this article.

For the FEADSBO, several deep learning-based features can be extracted from different system components. The potential features that can be extracted from each component, include: Received Signal Strength (RSS) Channel State Information (CSI) Carrier Frequency Offset (CFO) Amplitude & Phase Offset Signal-to-Noise Ratio (SNR) Antenna Array Configuration ADC and DAC Performance Metrics Energy Consumption Metrics. The proposed FEADSBO utilizes decentralized FL (DFL) and centralized FL (CFL) to achieve fast convergence by exploiting node-to-node communication. The feature extraction is performed at each node and the local model is trained on these features. This established a local level of intelligent synchronization. The local model are shared from node to node for DFL and simultaneously local model are shared with a centralized server for the global model in generation. The architecture of the proposed way-forward is depicted in Fig. 3.

## OPEN RESEARCH ISSUES

FL enabled 6G era is still an emerging area, and there is much to explore regarding the security analysis of this joint technological ecosystem. The following are pertinent open research issues on the way to achieving secure FL enabled 6G.

### FL for high-frequency mmWave communication

The extremely high-frequency radio waves are utterly useful in IAB propagation links and thus can contribute to the backhaul packet data transmission in either standalone (SA) or

no-standalone (NSA) modes. Nonetheless, with the simultaneous operation of the mm Wave and mmWave spectrums and the M-MIMO operation, the beam width has an inverse relationship with the number of directions to be scanned at the network setup time (*Qamar et al., 2019*). Narrower directional beams with analog BF and sequential scans can verily cost initial access delay. Also, a very small size wavelength can easily create hardware composition problems and thus demand advancement in the hardware equipment to make it ultra-reliable with massive antenna nodes for both access and backhaul links (*You et al., 2020*). The mmWave communication is a promising technology for future B5G/6G mobile communication services. The massive leap in wireless devices, 3D applications, immersive multimedia activities, and various Internet-based services will insist on the different antenna configurations and access to spectrum above 300 GHz. The collaborative work of mmWave with ultra M-MIMO antenna elements would enable the management of network resources, different controlling functional parameters, and network operations affluently (*Borges et al., 2021*). Yet, the close packet ultra M-MIMO deployment would complicate signal processing and estimation; therefore, an FL-based assessment with minimum complexity and computing time is needed to support 6G applications.

## FL for energy efficiency in massive antenna system

Severe path loss is an inevitable impediment to mmWave communication. With M-MIMO deployment, we can satisfactorily manage the problem at the high cost of EE and hardware complications. To surmount the poor energy management, a new array design has been developed wherein the mutual coupling between closely placed antennas is exploited to form super directive pairs (*Borges et al., 2021*). Thereby, the effectiveness of the proposed scheme is shown in three ways, minimize the number of antennas at the BS, increase EE, and ensure an achievable throughput rate. Principally, signaling cycles are adopted to construct the carrier leaner under low user mobility or traffic load in a cell(s). The NR 5G enables a longer sleep duration of up to 160 ms. Therefore, optimizing different parameters through FL integration mmWave concepts such as system information block 1, identification of cells, signaling pertinent to CSI, selection or reselection of signaling and process, and paging, are open research issues (*López-Pérez et al., 2021*).

## Secure integration of mmWave in FL enabled 6G networks

While mmWave communication offers the potential for high data rates and low latency, its susceptibility to atmospheric absorption and limited range raises concerns about the reliability and security of communication links (*Zhu et al., 2017*). In the context of FL-enabled 6G networks, where decentralized learning models are collaboratively trained across distributed devices, ensuring the confidentiality and integrity of model updates becomes critical (*Catak et al., 2022*). Optimum utilization of mmWave communication within FL-enabled 6G networks addresses the challenges related to secure key exchange, robust authentication mechanisms, and privacy-preserving protocols while safeguarding against potential security threats and vulnerabilities (*Kazmi et al., 2023a*). Research efforts are needed to develop efficient cryptographic solutions and communication protocols that accommodate the unique characteristics of mmWave channels, providing a foundation for

the reliable and secure integration of mmWave technology in future 6G networks employing FL.

## Mobility in mmWave with FL enabled 6G networks

6G communication leverages mmWave bands for enhanced data rates and capacity. However, the challenges associated with mobility management have become more pronounced due to the unique propagation characteristics of mmWave signals (*Qamar et al., 2017*). Addressing seamless handovers, efficient beamforming, and context-aware mobility policies in mmWave environments is essential. Furthermore, incorporating FL techniques introduces complexities in distributed learning across heterogeneous devices while ensuring privacy and security (*Fernandes et al., 2021*). Balancing these aspects to design a robust and adaptive mobility management framework that optimally exploits mmWave frequencies while leveraging FL for intelligent decision-making, remains an intriguing challenge for researchers in the pursuit of realizing the full potential of 6G networks (*Shome, Waqar & Khan, 2022*). Thus, integrating mobility management in mmWave frequencies, coupled with FL mechanisms in 6G, represents a critical open research issue.

## CONCLUSION

The integration of FL into 6G networks, particularly in the context of mmWave communications, holds immense potential for realizing the promises of ultra-high data rates and unprecedented connectivity. The unique propagation characteristics and security challenges associated with mmWave increase the complexity of effectively utilizing this spectrum. Therefore, in the integrated paradigm of mmWave and 6G network, FL emerges as a promising solution by enabling collaborative model training while ensuring data privacy. This comprehensive review has delved into the concepts of mmWave communications within the context of FL-enabled 6G networks, identifying challenges such as bandwidth consumption, power consumption, and synchronization requirements. By critically analyzing identified challenges, this study also suggests a FL based way forward for integrated scenarios of mmWave and FL. Thereby, this review has highlighted potential open research issues. The insights presented in this study pave the way for a transformative synergy between the mmWave spectrum and FL enabled 6G networks.

### Funding

This work was supported by the Universiti Kebangsaan Malaysia Fundamental Research Grant Scheme (FRGS) from the Ministry of Higher Education with the code: FRGS/1/2022/ICT11/UKM/02/1 and FRGS/1/2023/ICT07/UKM/02/1. The funders had no role in study design, data collection and analysis, decision to publish, or preparation of the manuscript.

## Grant Disclosures

The following grant information was disclosed by the authors:

Universiti Kebangsaan Malaysia Fundamental Research Grant Scheme (FRGS) from the Ministry of Higher Education: FRGS/1/2022/ICT11/UKM/02/1 and FRGS/1/2023/ICT07/UKM/02/1.

## Competing Interests

The authors declare that they have no competing interests.

## Author Contributions

- Faizan Qamar conceived and designed the experiments, performed the computation work, prepared figures and/or tables, and approved the final draft.
- Syed Hussain Ali Kazmi conceived and designed the experiments, performed the computation work, prepared figures and/or tables, authored or reviewed drafts of the article, and approved the final draft.
- Maraj Uddin Ahmed Siddiqui conceived and designed the experiments, performed the experiments, performed the computation work, prepared figures and/or tables, and approved the final draft.
- Rosilah Hassan performed the experiments, analyzed the data, authored or reviewed drafts of the article, and approved the final draft.
- Khairul Akram Zainol Ariffin analyzed the data, authored or reviewed drafts of the article, wrote the "Way forward" section, and approved the final draft.

## Data Availability

This is a literature review.

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
