# Peer review of "Federated learning for millimeter-wave spectrum in 6G networks: applications, challenges, way forward and open research issues"

_PeerJ Computer Science, doi:10.7717/peerj-cs.2360_

## Round 0.1 · original submission · Minor Revisions

All the authors have recommended some minor changes. These comments should be addressed

·

Basic reporting

No comment.

Experimental design

The authors have comprehensively surveyed the federated learning applications, challenges, and open research issues in 6G networks. The organiation is OK. Quotes and paraphrases are appropriate. However, the paper still has some issues in terms of inaccurate/confusing statements. For example, 1)in Section II, some equations are not numbered. 2)Title of Section II-B "FL IN SURLL OF 6G-->FL IN SURLLC OF 6G".
In addition, in Table II, the authors summarize relevant surveys from 2021 to 2023. In general, it is recommended to analyze the research in recent 5 years. Therefore, please add more content in Table II.

Validity of the findings

The conclusions identify unresolved questions and future directions clearly. I have no comment.

Reviewer 2 ·

Basic reporting

There are some typo errors in the paper. Please improve.

Experimental design

1. Figure 1 presents VOSviewer based on titles and keywords. I suggest you classify the keywords.

2. Authors suggested Federated Learning (FL) for 6G networks. I suggest you provide information on advantages and disadvantages or comparison of other methods such as Deep Learning and Machine Learning.

3. Authors highlighted the challenge of security concern in 6G network, however Section IV only highlights bandwidth consumption, energy consumption and synchronization requirement. I suggest to include explanation on security and privacy in FL.

Validity of the findings

No comment

Reviewer 3 ·

Basic reporting

The paper includes a significant review of the millimeter-wave spectrum in 6G networks, providing a concise description of its applications and an intuitive roadmap. I found it challenging to grasp the concept of mmWave. I recommend including a section explained in conceptual terms for non-specialists, ensuring this review is accessible across multidisciplinary audiences.

Experimental design

I found it challenging to grasp the concept of mmWave. I recommend including a section explained in conceptual terms for non-specialists, ensuring this review is accessible across multidisciplinary audiences.

Validity of the findings

Consistent methodology and intuitive flowcharts

Additional comments

The subject matter is of high scientific and technological impact, addressing potential uses of mmWave in 6G networks and the associated challenges. I suggest including just one chapter or paragraph explaining the Millimeter-Wave Spectrum conceptually for readers outside the field.

---

## Round 0.2 · accepted · Accept

The authors have answered the reviewers's comments appropriately.

Reviewer 2 ·

Basic reporting

The Author has addressed all the comments accordingly.

Experimental design

The Author has addressed all the comments accordingly.

Validity of the findings

The Author has addressed all the comments accordingly.